# Modulation of the Release of a Non-Interacting Low Solubility Drug from Chitosan Pellets Using Different Pellet Size, Composition and Numerical Optimization

**DOI:** 10.3390/pharmaceutics11040175

**Published:** 2019-04-10

**Authors:** Ioannis Partheniadis, Paraskevi Gkogkou, Nikolaos Kantiranis, Ioannis Nikolakakis

**Affiliations:** 1Department of Pharmaceutical Technology, School of Pharmacy, Faculty of Health Sciences, Aristotle University of Thessaloniki, 54124 Thessaloniki, Greece; ioanpart@pharm.auth.gr (I.P.); gkogkop@gmail.com (P.G.); 2Department of Mineralogy-Petrology-Economic Geology, School of Geology, Faculty of Sciences, Aristotle University of Thessaloniki, 54124 Thessaloniki, Greece; kantira@geo.auth.gr

**Keywords:** pellets 1, pellet diameter 2, crystallinity 3, sphericity 4, fast release 5, extended release 6

## Abstract

Two size classes of piroxicam (PXC) pellets (mini (380–550 μm) and conventional (700–1200 μm)) were prepared using extrusion/spheronization and medium viscosity chitosan (CHS). Mixture experimental design and numerical optimization were applied to distinguish formulations producing high sphericity pellets with fast or extended release. High CHS content required greater wetting liquid volume for pellet formation and the diameter decreased linearly with volume. Sphericity increased with CHS for low-to-medium drug content. Application of PXRD showed that the drug was a mixture of form II and I. Crystallinity decreased due to processing and was significant at 5% drug content. Raman spectroscopy showed no interactions. At pH 1.2, the dissolved CHS increased ‘apparent’ drug solubility up to 0.24 mg/mL while, at pH 5.6, the suspended CHS increased ‘apparent’ solubility to 0.16 mg/mL. Release at pH 1.2 was fast for formulations with intermediate CHS and drug levels. At pH 5.6, conventional pellets showed incomplete release while mini pellets with a CHS/drug ratio ≥2 and up to 21.25% drug, showed an extended release that was completed within 8 h. Numerical optimization provided optimal formulations for fast release at pH 1.2 with drug levels up to 40% as well as for extended release formulations with drug levels of 5% and 10%. The Weibull model described the release kinetics indicating complex or combined release (parameter ‘*b*’ > 0.75) for release at pH 1.2, and normal diffusion for the mini pellets at pH 5.6 (‘*b*’ from 0.63 to 0.73). The above results were attributed mainly to the different pellet sizes and the extensive dissolution/erosion of the gel matrix was observed at pH 1.2 but not at pH 5.6.

## 1. Introduction

Pharmaceutical pellets are multi-particulate drug delivery systems where the whole dose is divided into subunits with spherical shape and narrow particle size distribution within the range of 0.1 to 1.5 mm [1]. This multi-particulate presentation has several advantages over the tablet single-unit dosage form [2]. Pellets distribute uniformly in the gastrointestinal tract, which results in less variability in gastric emptying time, lower plasma level fluctuation, improved drug absorption, and a reduction of dose dumping. Additionally, their flowability enables processing of an automatic fast operation capsule and tableting machines, and their spherical shape makes them ideal for application of coatings [3,4,5].

For the preparation of pellets, extrusion/spheronization can be applied [6,7]. Product quality is controlled mainly by the composition while machine settings are less critical [8]. Microcrystalline cellulose (MCC) is usually part of the pellet base [9] due to its ability to retain large amounts of water in its structure, which provides elasto-plastic wet mass suitable for successful extrusion and good product quality [10,11,12]. However, in certain cases, incompatibilities of MCC have been reported because of its tendency to adsorb drugs on fibrils [13] and cause possible chemical interactions [14,15,16]. In addition, its use is obstructed by the prolonged and uncontrolled release of poorly soluble drugs [17]. An alternative is to replace part of the MCC with hydrophilic polymers, which aims to facilitate and control penetration of the aqueous dissolution medium into the pellet matrix by increasing hydrophilicity and swelling [18].

Chitosan (CHS) is a natural polysaccharide product of the deacetylation of chitin, which is a widely abundant polysaccharide. It dissolves in weakly acidic media by protonation of –NH2 groups, which forms a non-disintegrating gel matrix at high concentrations in water [19]. Its pharmaceutical importance as a functional excipient lies in its biocompatibility, biodegradability, and non-toxicity [20,21]. Additionally, it enhances the solubility of drugs, their permeation through the gastric mucosa, aids gastric protection due to its potent cytoprotective and healing action in gastric ulcers, and acts as a sustain-release agent [22,23,24,25]. Its effect on dissolution is influenced by the degree of deacetylation, the viscosity grade, its content in the formulations, and the drug solubility [19,23,26,27]. At low contents, it may act as a disintegrant, but, at high contents, it forms a hydrophilic gel, which delays drug release [28,29,30]. Due to its function as a ‘molecular sponge’, it can be used as an alternative to MCC [31]. Piroxicam (PXC) is a non-steroidal, anti-inflammatory, anti-rheumatoid, and analgesic drug [32,33] assigned to Biopharmaceutical Classification System (BCS) Class II due to its poor solubility in water [34]. Since CHS is positively charged in acidic media due to the protonation of amine [35,36] and PXC is cationic (pKa 5.3, Reference [35]) and is not ionized in deionized water, chemical interactions between these two are not expected and any effects on release should be due mainly to the contribution of the individual components.

The aim of this work was to prepare different CHS/MCC/PXC pellet formulations of two size classes (mini and conventional) by extruding through screens with small (0.5 mm) or large (1.0 mm) openings. Different CHS viscosity grades were initially compared and the one that prompted greater drug solubility was selected. It was expected that, due to gel formation and diffusional release, a reduction of pellet size will improve the release rate, and, thus, avoid the need to add hydrophilic excipients such as lactose [30]. However, extrusion through small orifice screens may adversely affect pellet shape and, for this reason, optimal pellet formulations with high sphericity, flowability, and instant or extended release were elucidated by applying a mixture experimental design followed by numerical optimization. The effect of processing on drug crystallinity was also examined. Since piroxicam has pH-dependent solubility, release was tested in both acidic and deionized water [37,38]. The release mechanisms were explained by analyzing the data using the Weibull model, which presents a relatively newer kinetic approach utilizing the entire drug release profile. This provides a more thorough description of the release mechanism. This model was described originally for extended release solid forms by Bonferoni et al. (1998) and interpreted by Papadopoulou et al. (2006) [29,39].

## 2. Materials and Methods

### 2.1. Materials

Microcrystalline cellulose (MCC, Avicel^®^ PH-101, lot 6950C) was from FMC (Cork, Ireland) and chitosan (CHS) from Primex (Siglufjordur, Island). From the supplied CHS grades: TM 3493 (viscosity (η) 5 cps, deacetylation (DA) 90%), TM 3528 (η = 8 cps, DA = 96%), TM 3603 (η = 121 cps, DA = 90%), TM 3389 (η = 171 cps, DA = 95%), and TM 3425 (η = 463 cps, DA = 92%). The experimental CHS powders were prepared by mixing equal quantities of supplied TM 3493 with TM 3528 to give low viscosity experimental grade CHS1 (5–8 cps, DA = 93%), and TM 3603 with TM 3389 to give the medium viscosity experimental grade CHS2 (121–171 cps, DA = 92.5%). The high viscosity grade TM 3425 (463 cps, DA = 92%) was used as received and is denoted as the experimental grade CHS3. Chemo Iberica S.A., Spain supplied Piroxicam (PXC). Polyvinylpyrrolidone (PVP) (K25, wt~21000) was gifted from BASF (Ludwigshafen, Germany).

### 2.2. Solubility of PXC in pH 1.2 and 5.6 in the Presence of CHS

To determine drug solubility in 0.1 N HCl, solutions with 0.01%, 0.05%, and 0.1% *w*/*v* CHS were prepared in 50 mL 0.1 N HCl and excess drug (100 mg) was added to each. A saturated drug solution in 0.1 N HCl was also prepared for comparison. The solutions were kept at 37 °C for 24 h under agitation and, prior to analysis, they were centrifuged for 10 min at 4500 rpm (Labofuge 400R, Heraeus, Germany). The UV absorbance of the supernatant was measured at 334 nm (Pharma Spec UV-1700 Shimadzu, Kyoto, Japan) and converted to a concentration (mg/mL) from a reference curve [C = (Abs + 0.0085)/0.7437]. To measure solubility in deionized water (pH 5.6), 100 mg CHS (non-dissolving in pH 5.6) were suspended in 10 mL deionized water and an excess drug was added. The drug-saturated solutions were kept in closed containers at 37 °C for 24 h under agitation and centrifuged. The absorbance of supernatant was measured at 359 nm and converted to a concentration from a reference curve [C = (Abs + 0.0141)/0.4964].

### 2.3. Preparation of Pellets

Thirty-gram batches of CHS/MCC/PXC were blended in a Turbula^®^ mixer (W.A. Bachofen, Muttenz, Switzerland) for 20 min and then transferred into a cylindrical vessel (0.8 L) fitted with a three-blade impeller. PVP 25 binder solution in water (7.5% *w*/*w*) was gradually added over 5 min to give 5% *w*/*w* PVP concentration in the final dry pellets. PVP was added to the binder liquid to improve the consistency of wet mass [40]. Any further wetting liquid required was added as deionized water. The wet mass was immediately processed in a radial extruder (Model 20, Caleva Process Solutions, Dorset, UK) that was operated at 25 rpm and fitted with a 1-mm orifice screen for the production of conventional pellets or a 0.5-mm orifice screen for the mini pellets (both screens had 1.75-mm thickness). The extrudate was immediately processed for 5 min in a spheronizer (Model 120, Caleva Process Solutions) fitted with a 12-cm diameter cross-hatch friction plate (0.8-mm depth grooves and pyramidal protrusions), operated at 1250 rpm and corresponding to 7.85 m/s peripheral velocity. The pellets were dried (40 °C, 12 h) in a tray oven with air circulation (Hereaus, Germany).

### 2.4. Characterization of Unprocessed Materials

#### 2.4.1. Particle Size

Particle size was determined using an image processing and an analysis system comprised of a microscope (Leitz Laborux S, Wetzlar, Germany), a video camera (VC-2512, Sanyo Electric, Osaka, Japan), and software (Quantimet 500, Cambridge, UK). Powder samples dispersed in liquid paraffin were examined at 40× total magnification. A mean particle diameter was expressed as an equivalent circle diameter (diameter of a sphere with the same projected area as the particle).

#### 2.4.2. Pycnometric Density

Helium pycnometry was applied (Ultrapycnometer 1000, Quantachrome Instruments, Boynton Beach, Florida, FL, USA). The instrument was calibrated using a standard 7.0699 cm^3^ steel ball. Samples were accurately weighed (3 decimals) and purged for 10 min before measurement. Sample volume (average of 10 runs) was measured from the displaced gas. Measurements were taken in triplicate and mean values and standard deviations were calculated.

#### 2.4.3. Moisture Content

Samples of about 1 g were placed in an infrared radiation balance and heated at 105 °C (Halogen Moisture Analyzer HR73, Metler Toledo, OH, USA). Sample weight was automatically taken every 30 s and the process ended when the loss between two successive values was less than 0.01%. Moisture content (MC%) was expressed as the weight difference relative to the initial weight.

### 2.5. Characterization of Pellets

#### 2.5.1. Size, Shape, and Density

Pellet size and shape were determined using an image processing and analysis system, as previously described [41]. Mean pellet diameter was expressed as an equivalent circle diameter and shape as the index e_R_ (Equation (1)), which is sensitive to surface irregularity and pellet geometry, and its value increases with sphericity, which nearly reaches 0.75 for perfect spheres [42]. Examples of pellet shapes with corresponding e_R_ values are shown in Figure 1.
e_R_ = (2 × π × radius)/perimeter − √(1 − (width/length)^2^)(1)

The pycnometric density of the pellets was determined as described above for the unprocessed materials (Section 2.4.2).

#### 2.5.2. Packing Ability, Flowability, and Porosity

Packing densities (bulk and tapped after 300 taps) were determined with a tester fitted with a 25 mL cylinder (14 mm drop, Erweka SVM 101, USP1, Heusenstamm, Germany). Carr’s compressibility index (CC%), equal to the volumetric change relative to tapped volume, was calculated from the density values as the index of packing ability. Flowability was estimated with an apparatus constructed according to United States Pharmacopeia [43]. Samples were transferred into a cylinder of 1.5 cm internal diameter and a 5-mm orifice at the center of its base. Pellets flowing through the orifice were collected on the platform of a balance (Bel Engineering MARK330, Monza, Italy) located 5 cm underneath the cylinder. Weight data was recorded every 0.2 s and transferred to a computer via an RS-232 interface. Pellet porosity was expressed as ε% = [1 − (pellet density/powder density)].

#### 2.5.3. Physicochemical Characterization

##### Raman Spectroscopy

Raman spectra were recorded to detect possible interactions between the drug and excipients using a bench top Raman spectrometer (Agility, dual band 785/1064 nm model, BaySpec, San Jose, CA, USA) and supporting software (Agile 20/20). Unprocessed powders or pellet samples were placed in standard glass vials and scanned over the range of 100 to 2700 cm^−1^ of Raman Shift using the laser excitation line 785 nm, exposure time of 1 s, and a power of incident laser beam of 150 mW. The recorded spectra were the average of 100 runs.

##### Powder X-ray Diffraction (PXRD)

Changes in the crystalline state and crystallographic characteristics of the drug were examined using PXRD (PHILIPS PW1710 diffractometer with CuKα, Ni-filtered 1.5418 Å radiation wavelength, Phillips, Eindhoven, The Netherlands). The samples were scanned over the 3°–43° 2θ range, at a speed of 1.2/min. Identification of the crystalline phases was made by comparing with published PXRD data based on the appearance and intensity of the reflections. Crystallinity was quantified as the crystallinity index (CI%) expressed by the ratio of the intensity of the strongest reflectance of the drug in the pellets at 26° 2θ, relative to that of the pure drug at the same 2θ. The crystallinity loss (LC%) due to processing was obtained as a percentage of the difference between the CI% of drug in physical mixtures (PM) and in pellets relative to that in PM (see Equation (2)).
LC% = [(CI% of drug in PM − CI% of drug in pellets)/CI% of drug in PM] × 100(2)

### 2.6. In-Vitro Release

In-vitro release of PXC was tested using the USP II Apparatus at 100 rpm and 37 ± 0.5 °C. Pellet samples of the experimental batches corresponding to 20 mg of drugs were added into 900 mL dissolution fluid. Since PXC has pH-dependent solubility, tests were conducted in two media including HCL 0.1 N (pH 1.2) and deionized water (pH 5.6). In the last case, pH was measured at the beginning and the end of the test and no significant change was recorded. Pellets of MCC/PXC without chitosan were also tested for comparison. Aliquots were taken at timely intervals and analyzed by UV spectroscopy (Pharma Spec UV-1700 Shimadzu, Kyoto, Japan) at 334 nm for pH 1.2 and 359 nm for pH 5.6.

#### Kinetic Models

The drug release data were analyzed using the Weibull equation [44]
ln[−ln(1 − *W*/*W*o)] = −ln*a* + *b*ln(*t* − *t*_o_)(3)
where *W* is the drug released at time *t*, *W*o is the drug released at the end of the test, *t*_o_ is the lag time before release as determined by trial and error for best line fitting, ‘*b’* is the constant characteristic of the shape of the release curve and the release mechanism [39], and a is a time-scale parameter defined as *a* = (td)*^b^* where td is the time required for 63.2% release.

### 2.7. Experimental Design and Optimization of Compositions for Instant or Extended Release

The influence of composition on the properties of pellets and drug release was studied separately for mini and conventional pellets. This doubled the number of experimental batches and, for this reason, the d-optimal mixture design including vertices, edge centers, centroid, and axial points was used as an efficient design applicable to constrained regions [45] (Table 1). The sum of components CHS (X1), MCC (X2), and PXC (X3) was 95% and the remaining 5% was PVP. Constraints were applied for CHS and MCC at 10% < X1, X2 < 80%, and for the drug at 5% < X3 < 70%, which provides a realistic design space. Consequently, the % weights were transformed into a ‘real’ scale (Equation (4)), where their sum is 1.0, and then into L-pseudo levels (Equation (5)) where their minimum value is 0 and the maximum is 1.
Real = Actual/Total of Actuals(4)
Pseudo = (Real − Li)/(1 − L)(5)
where Li is the lower constraint, and L is the sum of lower constraints.

Multiple linear regression analysis (backward elimination) based on Scheffé polynomial (Equation (6)) was used to derive model equations between the component levels and the following pellet properties: mean diameter, shape index, dissolution efficiency at pH 1.2, and drug release at pH 5.6 in 2 h and 8 h. The model includes linear and interaction terms.
Y = b1X1 + b2X2 + b3X3 + b12X1X2 + b13X1X3 + b23X2X3(6)

Contour and trace plots were constructed from the regression equations. Contour plots show the effect of composition on a property. Each component line of the trace plots describes the effect of that component by moving along an imaginary straight path connecting the centroid of the experimental design (intersection of the three lines in the plots) to the vertex of that component in the triangle. This visualizes the effects of one single component while holding the ratio of the others constant. *p* < 0.05 was the statistically significant level, and R^2^ and adjusted Radj^2^ indicate goodness of fit.

The derived regression equations were subsequently used to optimize formulations for good sphericity (e_R_ in the range of 0.3–0.51), maximum flowability, and fast or extended release. The criteria for fast release were set as: maximum DE% and minimum td (time for 63.2% release, Weibull equation) for release at pH 1.2, and, for extended release, they were set as: minimum DE% for release at pH 1.2, which is less than 60% release in 2 h and a maximum after 8 h release at a pH of 5.6. Optimization was applied for preset drug levels. The respective equations for minimization and maximization of a response di were:di = (Ymax − Yi)/(Ymax − Ymin)(7)
di = (Yi − Ymin)/(Ymax − Ymin)(8)
where di is the desirability function of a response ranging from 0 to 1, Ymin is the lowest measured value, Ymax is the highest measured value, and Yi is any value. di = 0 if the response value is outside the desired range and di = 1 if the response value is within the desired range. For minimization or maximization, the di varies from 0 to 1. The overall desirability D (Equation (9)) denotes the geometric mean of the individual desirability of n responses.
D = (d1 × d2 × d3 ×… × dn)1/n(9)

Design Expert 8.0 (Stat-Ease, Minneapolis, MN, USA) was used to generate the experimental design, for statistical analysis, to draw the contour and trace plots, and for numerical optimization.

## 3. Results and Discussion

The effect of chitosan grade viscosity on the solubility of piroxicam in acidic pH 1.2 and deionized water was initially examined to select the grade that prompted greater ‘apparent’ solubility. The term ‘apparent’ is used to distinguish from the thermodynamic equilibrium solubility.

### 3.1. PXC Solubility and Influence of Chitosan

Figure 2 presents the ‘apparent’ solubility of piroxicam in water at both acidic pH 1.2 and deionized water (pH 5.6) measured in the presence of different CHS viscosity grades. In Figure 2a, the ‘apparent’ solubility of PXC at pH 1.2 (where CHS dissolves) was plotted against CHS concentration while the bars in Figure 2b show ‘apparent’ solubility at pH 5.6 in the presence of suspended CHS. Due to its primarily acidic character, PXC is more soluble in basic environments [35,36]. Its measured greater solubility in CHS-free water at pH 1.2 than at pH 5.6 (0.10, Figure 2a, compared with 0.023 mg/mL, Figure 2a,b) is attributed to its amphiphilic character that allows some ionization at pH 1.2 (protonation of -NH2) but not at pH 5.6. From Figure 2a, it can be seen that, at pH 1.2, the ‘apparent’ solubility increased remarkably from 0.10 mg/mL in CHS-free water to 0.165 mg/mL in the 0.01 *w*/*w* CHS solution. Thereafter, the increase differed for each grade. For the low viscosity CHS1, it was small and reached 0.194 mg/mL at the highest 0.1% *w*/*w* CHS while, for the medium CHS2 and high CHS3 viscosity grades, it increased exponentially. It reached 0.238 and 0.222 mg/mL. Similarly, Figure 2b shows that the presence of CHS as suspended polymer in deionized water also increased drug solubility, but, to a lesser extent than pH 1.2, and the differences between the drug solubilities in the three CHS suspensions were small (0.138, 0.157, and 0.150 mg/mL for CHS1, CHS2, and CHS3, respectively). 

The greater effect of CHS on ‘apparent’ drug solubility at pH 1.2 compared to pH 5.6 is attributed to the solubility of CHS at pH 1.2 but not in deionized water (pH 5.6). At pH 5.6, polymer fibers sorb water but do not dissolve, and, hence, only those drug molecules adsorbed onto fibers benefit from wetting, which results in limited increased solubility. On the other hand, in the acidic environment, the dissolved CHS chains form water-soluble units with the drug molecules attached to the chains, which considerably increases ‘apparent’ drug solubility [46]. A possible explanation for the formation of chitosan/drug water soluble units could be weak hydrogen bonding between hydroxyl groups present in the chitosan with the hydroxyl of the protonated in the acidic pH zwitterionic tautomer of piroxicam [47]. The greater drug solubility improvement shown by the high viscosity CHS grades may be due to network formation of the CHS chains where the CHS-PXC water-soluble units are stabilized. The chains of low viscosity CHS are unable to form networks due to their mobility. Therefore, the CHS2 grade that prompted higher solubility was chosen to develop CHS/MCC/PXC pellets.

### 3.2. Technological Characteristics

#### 3.2.1. Wetting Liquid Volume, Pellet Size, and Shape

The measured properties of the experimental materials are given in Table 2. From Table 2, it can be seen that the experimental powders had similar median (d50) particle diameters for CHS2 21 μm, for PXC 11 μm, and MCC 31 μm, and similar pycnometric densities for CHS2 1.64 g/cc, MCC 1.69 g/cc, and PXC 1.58 g/cc. The similarity in particle diameters and densities favors good mixing and homogeneous paste formation [48,49]. The moisture contents of the unprocessed materials fell within the expected ranges.

Table 3 presents the volumes of added liquid and the paste consistencies that, after extrusion/spheronization, gave the highest sphericity for mini and conventional pellets with a mean pellet diameter and shape indexes. Batches B, F, and H with high CHS2 (80%, 45%, and 54.37%) showed high consumption. Although both CHS and MCC consist of fibrils in a sponge-like structure, chitosan-rich pellets, e.g., batch B with 80% CHS2, consumed larger amounts of liquid than MCC-rich pellets, e.g., batch G with 80% MCC (58.0 mL compared to 42.0 mL). However, the wet mass of batch B had low plasto-elasticity, that separated under light pressure, and gave low sphericity pellets after spheronization (e_R_ 0.27 and 0.32 for the mini and conventional pellets, respectively, Table 3) with a large proportion of small-sized pellets, as reflected by the low mean diameters (367 and 712 μm, Table 3). This indicates that only part of the consumed liquid was used to form a mass suitable for extrusion, with the remainder residing loosely in the interior [31]. Batch D with the highest drug content (70%) gave a creamy paste despite a low consumption value (21 mL), which shows an inability to take up liquid and form an extrudable paste. This was also indicated by the low pellet sphericity (e_R_ 0.17 and 0.15 for mini and conventional pellets).

From Table 3, it can be seen that the mean diameters of the mini pellets were about half the value of the conventional pellets (367 to 585 μm compared to 712 to 1206 μm), as expected. Batches B, F, and H with high CHS2 content and greater liquid consumption gave smaller pellets than batches A, C, and G with high MCC (mean diameters of 367, 411, and 384 μm when compared to 532, 516, and 489 μm for the mini pellets, and 712, 984, 967 μm when compared to 1010, 1013, and 1021 μm for the conventional pellets, respectively). The relationship between liquid consumption and pellet diameter (not shown graphically) was linear and inversely proportional (R^2^ 0.689 for the mini and 0.820 for the conventional pellets), which generally agrees with previous reports [26]. Mini pellets were less spherical (e_R_ range 0.15–0.44 compared to 0.16–0.51, Table 3), and this can be attributed to the more efficient absorption of centrifugal and frictional forces exerted during spheronization by the conventional pellets due to their larger size and mass.

#### 3.2.2. Packing, Flowability, and Porosity

Table 4 presents pycnometric (Ps) and packing densities (bulk Pb and tap Pt), porosity (ε%), Carr’s compressibility index (CC%), and flowability. Tap density expresses the extent while CC% expresses the easiness of packing. The properties of the mini and conventional pellets were correlated linearly (R^2^ for: Ps = 0.856, Pb = 0.818, Pt = 0.815, ε% = 0.939, and flowability of 0.619). Mini pellets showed greater Pt values (from 0.67 to 0.87) compared to the conventional pellets (0.57 to 0.79), which is attributed to their smaller size, and, therefore, smaller inter-particle voids less occupied volume. Batches B and D with irregular pellet shape gave the lowest p_t_ values. The parameter CC% presented low values (<15%) for both mini and conventional pellets, which indicates good packing ability.

The porosity ranges of the mini and conventional pellets were similar (0.31–12.45% and 0.20–14.10%, respectively). Batches B, E, and H with high CHS2 and up to 21.25% PXC showed low ε% (6.53%, 8.19% and 8.47% for the mini and 5.35%, 5.75 and 8.27% for the conventional pellets), which implies a denser structure that is attributed to the binder action of CHS [40]. However, while batches A, F, and G with high MCC showed higher ε% values (10.14%, 11.48%, and 12.45% for mini pellets and 12.13%, 12.30%, and 14.10% for conventional pellets), which is similar to those previously reported for MCC pellets [50]. Batches C, D, and I with high PXC showed unusually low porosities (4.29%, 0.31%, and 3.89% for the mini and 1.97%, 0.20%, and 3.08% for the conventional pellets).

Flowability depends on pellet size, shape, and orifice diameter. The 6-mm orifice was more than six times larger than the diameter of the pellets (means of 452 μm and 992 μm for mini and conventional, respectively, Table 3) and was selected to avoid blocking [51]. From Table 4, it can be seen that the flowability of the mini pellets was considerably greater than conventional pellet flowability (from 1.76 to 2.82 g/min when compared to the 1.35 to 1.99 g/min) despite their lower overall sphericity, which is due to their smaller diameter and easier movement through the orifice.

#### 3.2.3. Analysis of Mixture Design for the Effect of Composition on Pellet Size, Shape, and Flowability

Statistically significant model equations with fitting indices derived from the analysis of the experimental design that describe the effects of composition on the pellet diameter, the shape, and flowability for the mini and conventional pellets are presented in Table 5. In all cases there was no lack of fit. Respective contour and trace plots are presented in Figure 3, Figure 4 and Figure 5. The plots in Figure 3 show that increasing CHS2 decreased the mean pellet diameter for both mini and conventional pellets while PXC caused a diameter increase and MCC had a minimal effect. The model equations were linear (R^2^ 0.837 and 0.742 for mini and conventional pellets respectively). The decrease observed at high CHS2 content is associated with high wetting liquid consumption followed by shrinkage during drying. The increase of the pellet diameter at high PXC is due to the irregular, elongated pellet shape (Figure 1b).

Considering the pellet shape, from the trace plots in Figure 4, it can be seen that sphericity was adversely affected by PXC for both mini and conventional pellets. It was positively affected by MCC while CHS2 had a small effect. The increased irregularity at high drug content is due to the lack of sufficient excipients resulting in a creamy paste even at low volumes of added liquid (batch D, Table 3). The regression model for both mini and conventional pellets was reduced quadratic (R^2^ 0.885 and 0.901) with an interaction of the effects of MCC and PXC, i.e., at low MCC levels, the effect of drug content was small but strongly negative at high MCC (Figure 4b). Furthermore, from Figure 5, it can be seen that MCC had a positive effect, whereas the drug had a negative effect on flowability, which can be explained by their respective effects on sphericity (Figure 4). The model describing the effects of composition on flowability was reduced quadratic for both mini and conventional pellets (R^2^ 0.838 and 0.956), which was indicated by curvilinear plots.

### 3.3. Physicochemical Evaluation

#### 3.3.1. Raman Spectroscopy

Raman spectra provide information regarding interactions of drugs with excipients. In Figure 6, the spectra of chitosan (CHS2), microcrystalline cellulose (MCC), piroxicam (PXC), and batches C, l with 37.5% drug and low or high CHS/MCC ratio (10/47.5 and 47.5/10), respectively, are presented. The CHS2 spectrum showed two small peaks at 119 cm^−1^ and 583 cm^−1^. The MCC spectrum showed one small double peak at 1196/1120 cm^−1^. The PXC spectrum showed a large fingerprint region in the range of 1090 to 1661 cm^−1^ (enclosed by vertical dotted lines), which is due to the complex patterns of C–C, C–N, and aromatic ring vibrations [52]. The drug peaks at 1543 and 1570 cm^−1^ are due to ring stretching and C=C symmetric stitching vibrations, respectively (indicated by arrows), are present in both C, I spectra and are characteristic of form II [53]. The spectra of both C and I showed the same peaks in the fingerprint region as the drug spectrum, which indicates no polymorphic transformation or interaction due to processing. Furthermore, peaks at 1007 and 1401 cm^−1^ that are characteristic of the PXC monohydrate [35,53] are not seen in the spectra of drug or pellet batches, which indicates an absence of the monohydrate drug.

#### 3.3.2. Powder X-ray Diffraction (PXRD)

PXRD patterns provide detailed information about the crystallinity of the drug in the pellet batches. Figure 7 presents the patterns of CHS2, MCC, and PXC powders, pellet batches B and G with low 5% drug and high (80%) or low (10%) CHS2, and batches H and A with higher 21.25% drug and high (54.37%) or low CHS2 (19.38%). The CHS2 pattern showed one small reflection at 10.9° 2θ and one strong reflection at 20.0° 2θ, whereas the MCC pattern showed three reflections with one broad between 15° and 16.5° 2θ, one strong at 22.3° 2θ, and one small at 34.2° 2θ.

The PXC pattern showed sharp reflections at 9.1° and 10.2° 2θ, four consecutive reflections between 15° and 17° 2-θ, and strong, sharp, reflections at 15.9°, 26.1°, and 27.1° 2θ. The strong reflection at 9.1° and the consecutive reflections at 26.1°/27.1° 2θ confirm the presence of the anhydrous form, which is in agreement with the Raman spectroscopy results. Additionally, from the PXRD pattern of PXC and using the database of the International Center of Diffraction Data (ICDD 2003) [54], it can be inferred that the form-II (44-1839 ICDD card) of the piroxicam structure is the major phase, while the form-I (40-1982 ICDD card) is identified as the minor phase. More specifically, using the mass absorption coefficient, the density, and the specific reflections of each form, it is estimated that form-II constitutes 69% *w*/*w* of the drug, while form-I is 31% *w*/*w*. From the patterns of the pellets shown in Figure 7, it appears that the drug reflections at 9.1°/10.2° and 26.1°/27.1° 2θ are still discernible in the PXRDs of batches G and B (though small due to the low drug content), and are clearly seen in the PXRDs of batches A and H with higher drug content, which indicates the predominantly crystalline state of the drug.

Crystallinity indices of the drug for the experimental pellet batches with 5% (B, G), 21.25% (A, H), and 37.5% drug (C, I), and their corresponding physical mixtures are presented in Table 6. In all cases, there is a loss of percentage in crystallinity due to processing, which is greater for the low drug batches B and G (21.14–22.57% compared to 2.51–3.87% for A and H, and 2.54–3.98% for C and I). It is documented that CHS2 is able to amorphize drugs processed by wet granulation [25]. (PVP is also an amorphizer but, because it is included with a low content of 5%, its contribution is negligible [55]). This effect should result from the dissolution of some drug content during wetting/extrusion, which is subsequently converted to an amorphized form after drying. Since the effect of CHS on solubility at a pH of 5.6 is independent of CHS content (Figure 1b), the amount of amorphized crystalline drug should not differ between batches, which explains the greater percentage of crystallinity loss observed in batches B and G with low drug content. Comparing batches with the same PXC content, it can be seen that the differences in crystallinity loss are small, regardless of the CHS/MCC ratio.

### 3.4. In-Vitro Release

In Figure 8, release profiles of mini and conventional pellets at two pH media (pH 1.2 and 5.6) are presented together with data from MCC/PXC pellets without chitosan for comparison purposes. Although the stay of the drug in the stomach delivered by conventional formulations is usually less than 120 min., this time may increase considerably for the present pellet formulations due to the muco-adhesive properties of chitosan [56]. For this reason, release studies of up to 500 min were conducted since the release exceeded this period for some pellet batches. Representative images of pellets before and after dissolution at pH 1.2 are shown in Figure 9 for batches A and C. Figure 10 presents images of all experimental batches before and after dissolution in deionized water (pH 5.6). The dissolution efficiency (DE%) and similarity factors (f2) comparing drug release from mini and conventional sized pellets at pH 1.2 and pH 5.6 are given in Table 7.

Comparing Figure 8a with Figure 8c–d, it can be seen that release at pH 1.2 is generally faster than at pH 5.6. This can be explained partly by the greater ‘apparent’ solubility of the drug at pH 1.2 than at pH 5.6 in the presence of CHS2 (0.238 compared to 0.157 mg/mL, Figure 2), but mainly due to the different state of the pellet matrix in the two pH media. Since CHS dissolves in pH 1.2 but not in pH 5.6, the pellet matrix dissolves and erodes during dissolution testing, in parallel to diffusion (Figure 9). However, in deionized water, erosion occurs to a much lower extent (Figure 10). Additionally, from Figure 8a–d, it can be seen that the pellet batch prepared with MCC/PXC only (black line open circles) showed much lower release, which underlines the importance of chitosan as already reported for other drugs [57].

From Figure 8a,b, it can be seen that, in six of nine cases, drug release at pH 1.2 completes within 3 h. Faster release was obtained from batches H and I with high CHS/medium or high drug (Table 1) (60–70% within 30 min from mini and about 50% from conventional pellets. Together with batch E, these batches also gave higher DE% values for both mini and conventional pellets (Table 7). Batches B and F with high CHS/low PXC and batch C with low CHS/high PXC, respectively, showed the lowest DE% (Table 7) and extended release. However, since erosion was involved (Figure 9), the extended release observed was highly dependent on composition and difficult to control. For example, the curves of batches B and C are located closer together than those of batches A and C (Figure 8a,b), despite the greater similarity in compositions of the latter two (Table 1).

The plateaus in the curves in the 8 h dissolution-testing period corresponded to less than 100% release, which may be partly ascribed to changes in solubility due to transformation into a hydrate form of lower solubility during dissolution [35], but also to the continued slow release after 8 h. In fact, measurements over a longer dissolution time showed that drug released (%) was still increasing and reaching above 85% after 24 h for both mini and conventional pellets, with the exception of conventional pellet batch B (70% release). For the purpose of this study, the end of the test period was considered to be 8 h. The DE% values for the mini and conventional pellets were linearly correlated (R^2^ = 0.880) and differences between their release profiles were small. Only batches B and H had similarity factors f2 < 50. These contained high CHS (80% and 54.37%, Table 1). Therefore, gel formation, diffusional release, and pellet size had higher significance than for other batches [30].

From Figure 8c,d, it can be seen that, at pH 5.6, the profiles of the mini pellets differed greatly from the conventional pellet profiles with the latter displaying much slower release (<50% after 8 h), which precludes practical use. From the values of the similarity factors f2 in Table 7, it appears that only batches A, C, D, and I showed similarity between mini and conventional pellets (f2 > 50). These batches all had high PXC content (37.5% or 70%) and exhibited erosion after dissolution testing (Figure 10), which alters the diffusional release. Figure 8c shows that the release curves of the mini pellets fall into three groups where: one group has a faster release (over 80% in 8 h) and comprises batches B and F, with high CSH2/PXC ratios of 16.0 and 9.0, respectively; a second group comprises batches G and H with CSH2/PXC ratios of 2.0 and 2.5 and releases 70–80% after 8 h; a the third group comprises batches A, D, E, and I with CSH2/PXC ratios of 0.91, 0.21, 1.25, and 1.27, respectively, showing low release between 50% and 60%. Batch C that had a CSH2/PXC ratio of 0.27, showed poor release. The latter five batches with reduced or poor release exhibited erosion and loss of matrix integrity after testing (Figure 10), which altered the beneficial release effect. On the other hand, batches B, F, G, and H of mini pellets, which did not disintegrate, showed good extended release over 8 h. Of these, batches B, F, and G also gave extended release at pH 1.2 and, therefore, appear to be promising extended-release formulations.

From the above, it appears that, when the CSH2/PXC ratio was ≥2, the pellet matrix retains the characteristics of the chitosan network, which results in extended and complete release after 8 h.

Table 5 shows statistically significant model equations derived from regression analysis of the experimental design describing the effect of composition on the dissolution efficiency (DE%) at pH 1.2 for mini and conventional pellets. The respective contour and trace plots are presented in Figure 11. From the contour plots, it appears that, at pH 1.2, greater DE% is obtained for intermediate levels of the three components, and the trace plots show clearly that CHS exerts the greatest effect. Statistically significant model equations for the effect of composition on DE% at pH 5.6 for mini and conventional pellets are also given in Table 5, with their respective contour and trace plots in Figure 12. It can be seen that, contrary to the effect of CHS2 at pH 1.2, at pH 5.6, greater DE% is obtained for the mini pellets at high CHS and low drug content and that these two components had the greatest effect (Figure 12). The conventional pellets showed very low DE% (values 9.40% to 31.49%, Table 7), which indicates poor release. The trace plots in Figure 12 show that for the conventional pellets at pH 5.6 the drug component had the greater effect on the release.

#### In-Vitro Release Mechanisms

Further elucidation of the release mechanisms is provided by examining the values of the ‘*b*’ parameter from the Weibull model, as presented in Table 8 for release at pH 1.2 and pH 5.6 for mini and conventional pellets. The generally high R^2^ values of the non-linear regressions indicate good model fitting. Considering the release times recorded at pH 1.2, it can be seen from Table 8 that, for the mini pellet batches A, B, C and G and the conventional pellet batches A, E, F, G, H, and I, the parameter ‘b’ is >1.0. This indicates a sigmoidal curve, i.e., a small initial increase up to the infection point (not visible in the curves since it occurs within in the first 15 min) and followed thereafter by an increase asymptotically to maximum.

Concerning release from the mini pellet batches at pH 5.6, the data in Table 8 show that, with the exception of batches G and D, ‘*b*’ parameter values ranged between 0.69 and 0.75, which indicates normal (Fickian) diffusion. Batch G presented *b* = 0.63, which indicates release by diffusion in a disordered matrix structure. This may be attributed to the high MCC or low CHS2 (10%) resulting in a non-homogenous structure and possible existence of ‘dry’ regions inside the gel during dissolution. Batch D showed *b* = 0.76, which indicates Fickian diffusion enhanced by a further release mechanism. This may be attributed to the high drug content and erosion during dissolution (Figure 10d). Regarding release at pH 5.6 for the conventional pellets, it can be seen from Table 8 that the ‘*b*’ values varied between 0.69 and 0.96, which implies operation of normal or combined diffusion. However, because less than 50% of the drug was released in the 8-h test period, a detailed analysis of the mechanisms involved would be unreliable and of limited value.

### 3.5. Optimization of Formulations

The numerical optimization was based on the modulation of the parameters e_R_ (>0.30) and flowability (maximize) for optimal technological properties and release as follows. For fast release in pH 1.2 solution, the DE% was maximized and the td was minimized. For extended release, DE% in pH 1.2 was minimized. Release within 2 h in pH 5.6 was set to 60%, and release within 8 h at pH 5.6 was maximized. As seen from the regression analysis results in Table 5, the above parameters were described by statistically significant models with high values of fitting indices. Therefore, reliable prediction and optimization of the formulations with optimal technological performance and fast or extended release can be computed. The results of the numerical optimization are presented in Table 9 as desirability values for fast (A) and extended release (B). Due to the poor release from the conventional size pellets at pH 5.6, optimization is presented only for release at pH 1.2.

Considering optimization at pH 1.2 for different preset drug levels, it can be seen that high values of desirability function above 0.919 (in a scale 0 to 1) were obtained for drug levels up to 40%. These represented pellets with acceptable shape (e_R_ > 0.30) and flowability (>2.15 g/min) that released drugs with dissolution efficiency between 63.97% and 77.60% for mini pellets and 54.33% to 73.05% for conventional pellets within a time-scale of 28.12 to 47.94 min and 40.73 to 52.51 min, respectively. Considering optimization at pH 5.6, it can be seen that good values of desirability function of 0.857 and 0.726 were obtained for pellet formulations with 5% or 10% drug content, with CHS2%/MCC%/PXC% compositions of 46.0/43.98/5.0 and 47.3/37.7/10.0, respectively (the analysis did not give meaningful results for drug levels exceeding 10%). These showed DE% values of less than 49.9% and 54% at pH 1.2 and pH 5.6, respectively, and released less than 60% and 53.1% within 2 h, and 93.3% and 85.4% after 8 h. Therefore, these formulations could be considered suitable for the further development of extended-release chitosan pellets of a poorly soluble non-interacting drug.

## 4. Conclusions

Medium viscosity chitosan grade prompted greater ‘apparent’ solubility improvement of piroxicam in both pH 1.2 and deionized water. However, due to the coherent gel consistency, drug release from conventional-sized pellets obtained with a 1-mm extrusion screen was extremely slow and excluded any possibility for practical use. On the contrary, use of mini pellets obtained with a 0.5 mm extrusion screen gave complete and extended release in deionized water following Fickian diffusion, as indicated by analysis of 8-h release data, according to the Weibull kinetic model. Batches of mini pellets prepared with a chitosan/microcrystalline cellulose ratio ≥2 and drug content up to 21.25% showed the best extended release that was nearly complete after 8 h and could be potentially used for further development. Optimization of the results of technological properties and drug release pointed out optimal formulations for fast-release mini pellets with CHS between 10.55% and 37.71%, MCC between 27.26% and 79.45%, and drugs up to 40%. For extended release, the optimal formulations were mini pellets with 5% drug, CHS2 = 46.02% and MCC% = 43.98%, and mini pellets with 10% drug, CHS2 = 47.32%, and MCC = 37.68%.

## Figures and Tables

**Figure 1 pharmaceutics-11-00175-f001:**
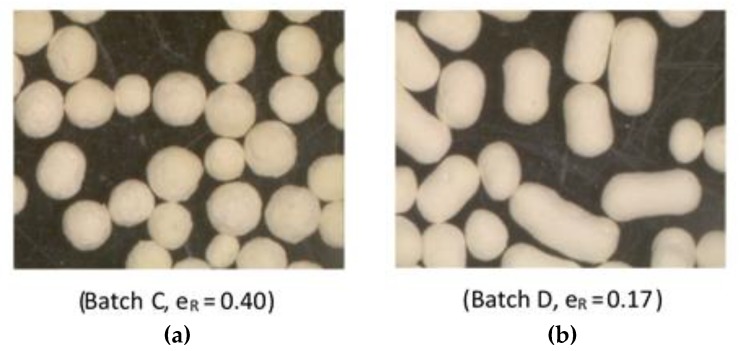
Images of pellet batches C (**a**) and D (**b**) showing different shapes.

**Figure 2 pharmaceutics-11-00175-f002:**
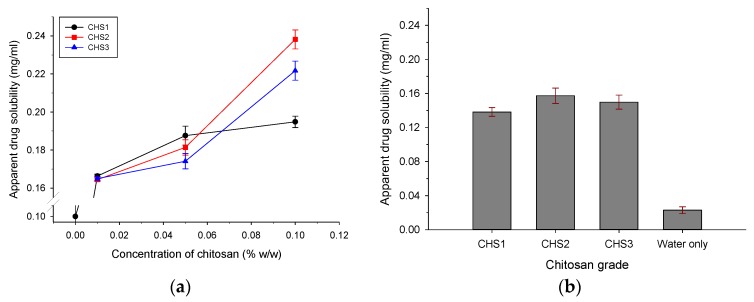
(**a**) Apparent solubility of piroxicam at pH 1.2 in the presence of increasing concentrations of low (CHS1), medium (CHS2), and high viscosity (CHS3) chitosan grades (error bars are standard deviations, *n* = 3). (**b**) Apparent solubility of piroxicam in deionized water (pH 5.6) in the presence of 100 mg chitosan suspended in 10 mL water.

**Figure 3 pharmaceutics-11-00175-f003:**
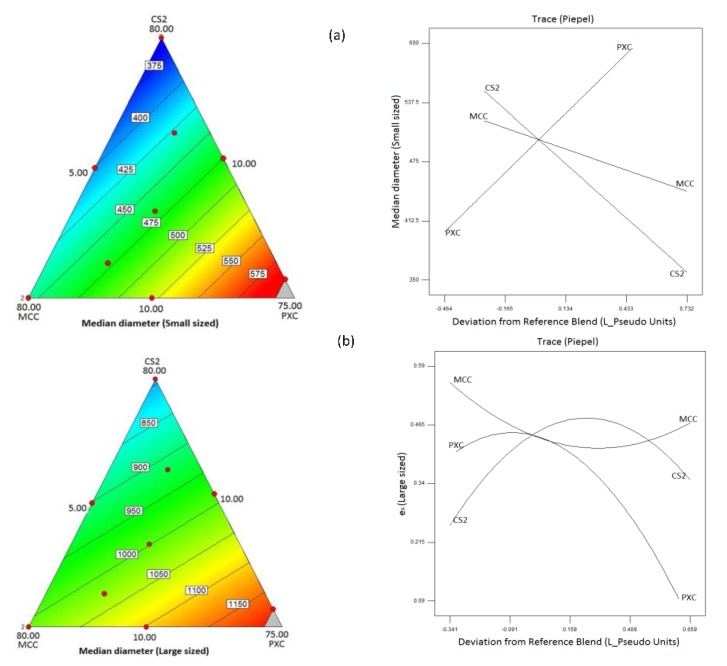
Contour and trace plots of mean pellet diameters for (**a**) mini and (**b**) conventional pellets.

**Figure 4 pharmaceutics-11-00175-f004:**
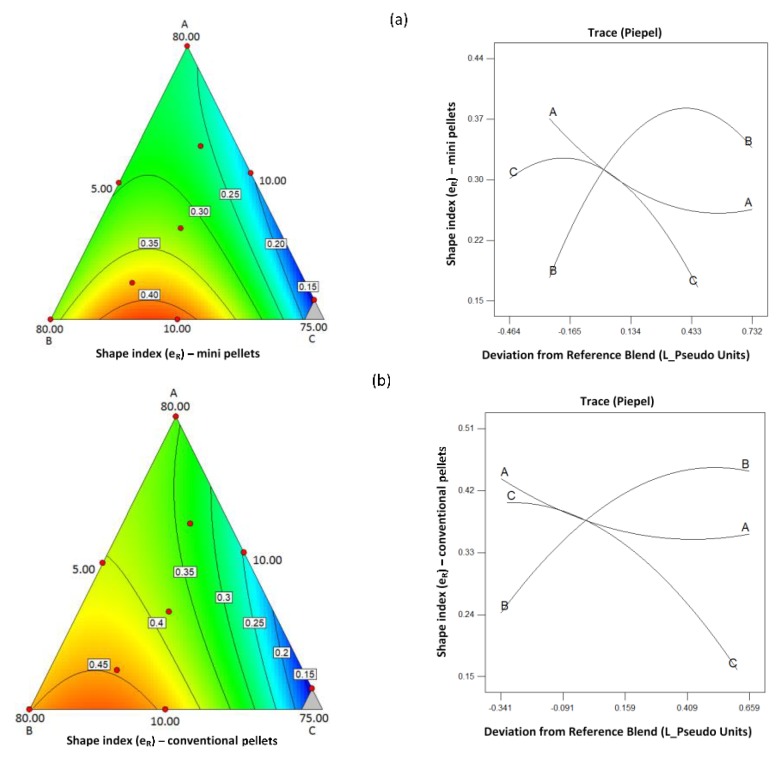
Contour and trace plots of the shape index e_R_ for (**a**) mini and (**b**) conventional pellets.

**Figure 5 pharmaceutics-11-00175-f005:**
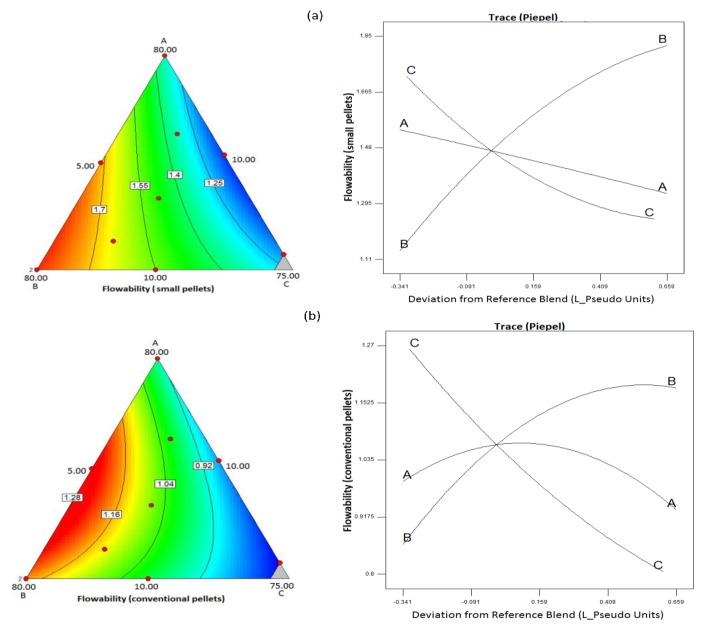
Contour and trace plots of flowability for (**a**) mini and (**b**) conventional pellets.

**Figure 6 pharmaceutics-11-00175-f006:**
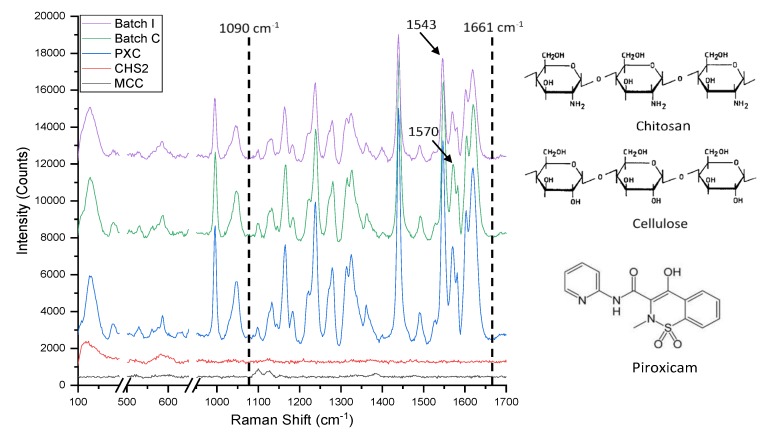
Chemical structures and Raman spectra of microcrystalline cellulose (MCC), chitosan (CHS2), unprocessed drug (PXC), and experimental batch I (CHS2 47.5%, MCCI 10%, PXC 37.5%).

**Figure 7 pharmaceutics-11-00175-f007:**
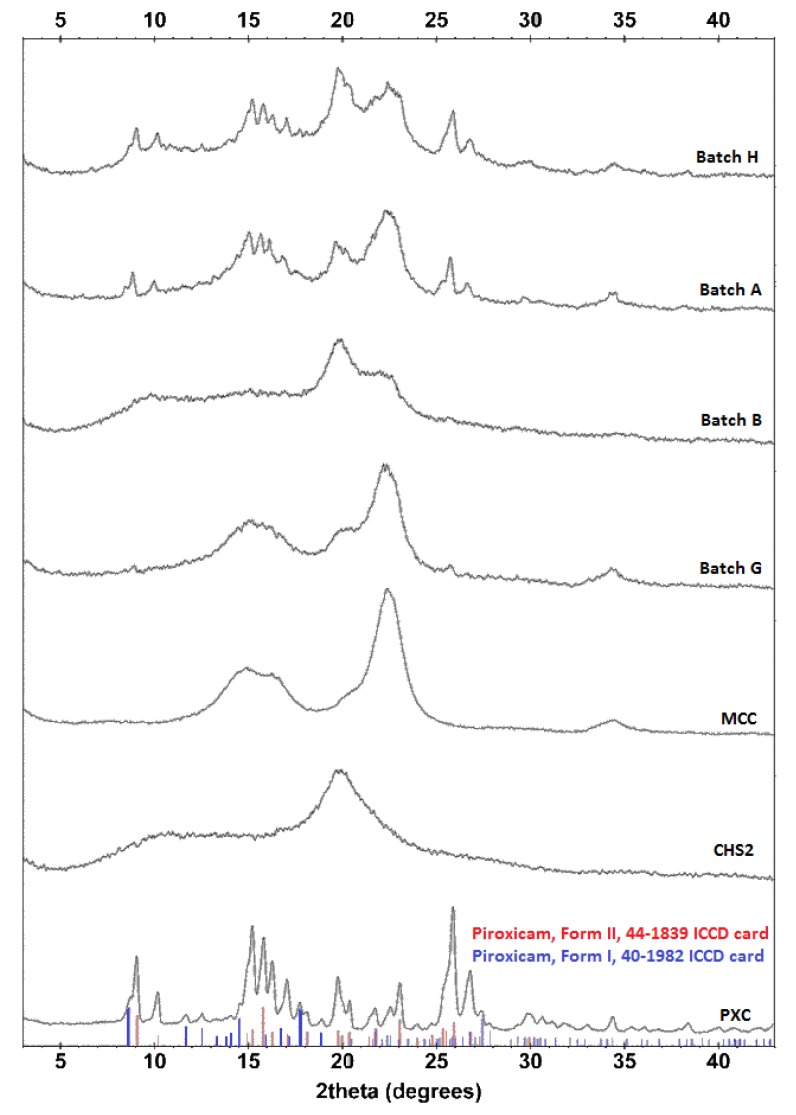
PXRDs for unprocessed materials and batches B and G with 5% drug content, and batches A and H with 21.25% drug content.

**Figure 8 pharmaceutics-11-00175-f008:**
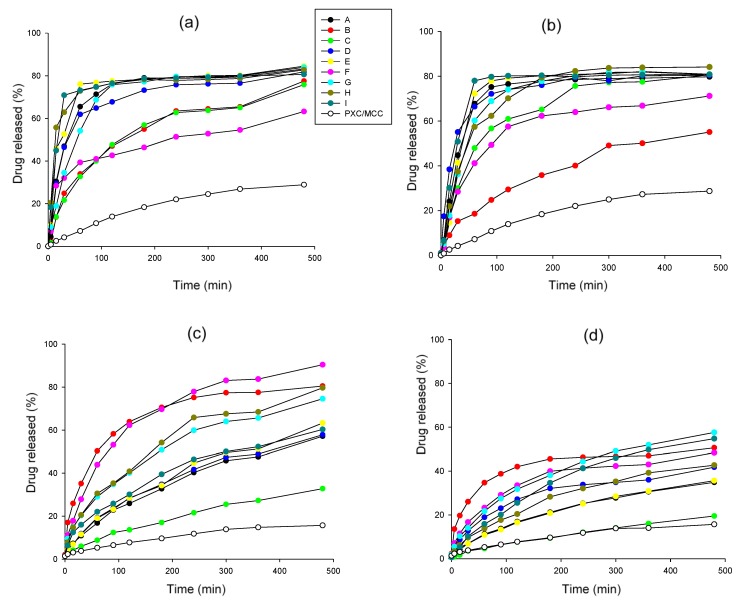
Release profiles of: (**a**) mini pellets at pH 1.2, (**b**) conventional pellets at pH 1.2, (**c**) mini pellets at 5.6, and (**d**) conventional pellets at pH 5.6. Release of drug from pellets prepared without chitosan (PXC/MCC) is also shown for comparison purposes.

**Figure 9 pharmaceutics-11-00175-f009:**
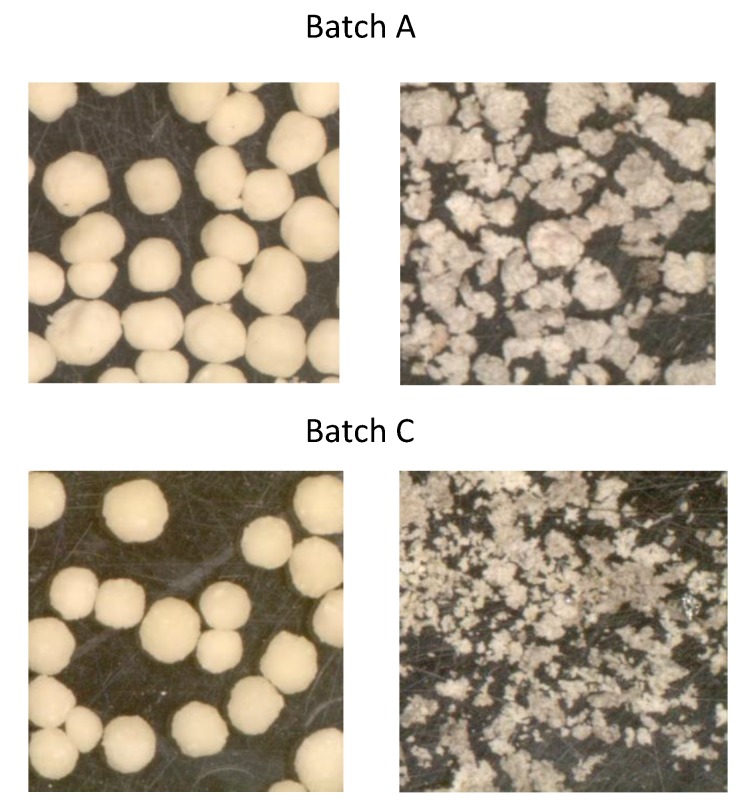
Images of pellets from batches A and C before (left) and after (right) the dissolution test at pH 1.2.

**Figure 10 pharmaceutics-11-00175-f010:**
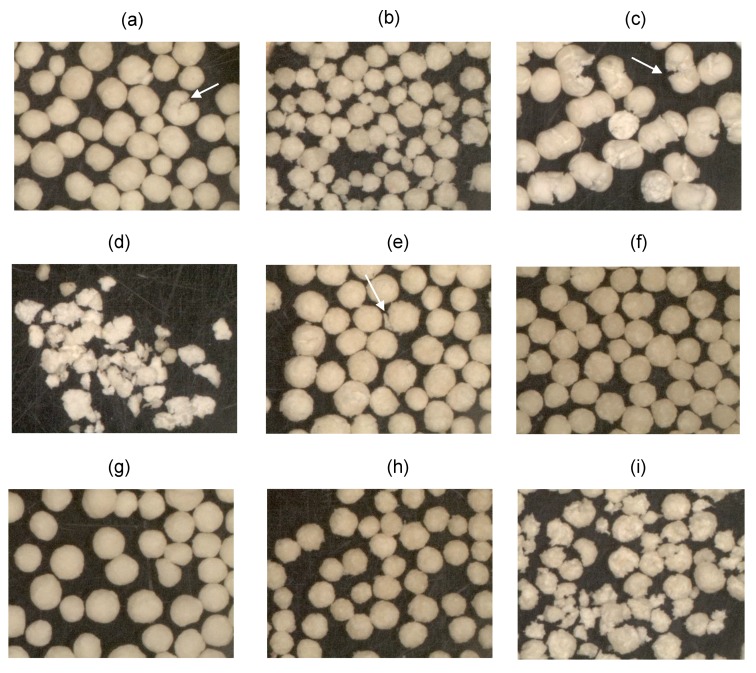
Images of pellets from all experimental batches (**a**–**i**) (Table 1) collected and dried after dissolution testing at pH 5.6 (Arrows indicate areas of disruption).

**Figure 11 pharmaceutics-11-00175-f011:**
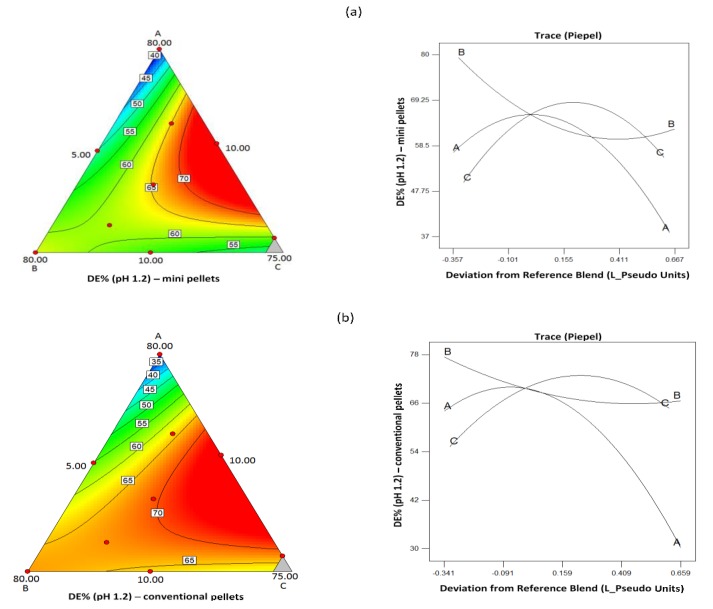
Contour and trace plots of dissolution efficiency for (**a**) mini and (**b**) conventional pellets at pH 1.2.

**Figure 12 pharmaceutics-11-00175-f012:**
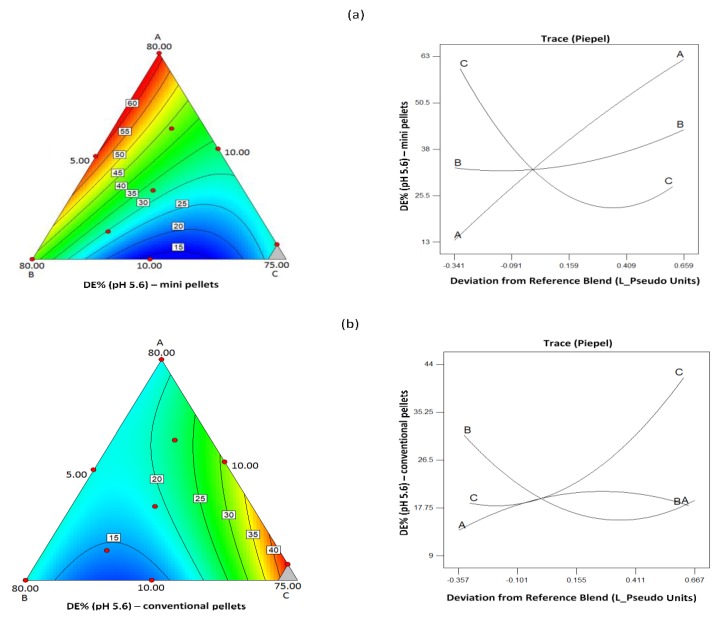
Contour and trace plots of dissolution efficiency for (**a**) mini and (**b**) conventional pellets at pH 5.6.

**Table 1 pharmaceutics-11-00175-t001:** Experimental mixture design for pellet batches. The design was applied to mini and conventional pellet batches separately.

Batch Code	Point in Design Space	Actual Values (%)			Real Values			L-Pseudo Values		
		CHS2	MCC	Drug	CHS2	MCC	Drug	CHS2	MCC	Drug
A	Axial	19.38	54.37	21.25	0.20	0.57	0.22	0.13	0.63	0.23
B	Vertex	80.00	10.00	5.00	0.84	0.11	0.05	1.00	0.00	0.00
C	Center edge	10.00	47.50	37.50	0.11	0.50	0.40	0.00	0.54	0.46
D	Vertex	15.00	10.00	70.00	0.16	0.11	0.74	0.07	0.00	0.93
E	Centroid	33.33	35.00	26.67	0.35	0.37	0.28	0.33	0.36	0.31
F	Center edge	45.00	45.00	5.00	0.47	0.47	0.05	0.50	0.50	0.00
G ^#^	Vertex	10.00	80.00	5.00	0.11	0.84	0.05	0.00	1.00	0.00
H	Axial	54.37	19.38	21.25	0.57	0.20	0.22	0.63	0.13	0.23
I	Center edge	47.50	10.00	37.50	0.50	0.11	0.40	0.54	0.00	0.46

^#^ This experimental batch was reproduced for testing lack of fit of the applied models.

**Table 2 pharmaceutics-11-00175-t002:** Properties of the unprocessed experimental materials.

Material	Moisture Content (%)	Particle Density (g/cc) ^#^	Particle Size Distribution Parameters ^#^ (μm)		
			d10	d50	d90
CHS2	8.78 ± 0.71	1.64 ± 0.01	10.3	21.0	83.2
MCC	5.38 ± 0.22	1.69 ± 0.01	10.7	31.0	100.6
PXC	0.32± 0.01	1.58 ± 0.01	6.4	11.0	28.6

^#^ d10, d50, and d90 diameters correspond to 10%, 50%, and 90% of the particle size distribution, respectively.

**Table 3 pharmaceutics-11-00175-t003:** Wetting liquid consumption, paste consistency, pellet size, and pellet shape of the experimental mini and conventional pellet batches.

Batch Code	Wetting Liquid (mL)	Paste Consistency	Mini Pellets		Conventional Pellets	
			Mean Pellet Diameter (μm)	Shape Index e_R_	Mean Pellet Diameter (μm)	Shape Index e_R_
A	40.0	Good	532	0.44	1010	0.39
B	58.0	Fragile	367	0.27	712	0.31
C	29.0	Good	516	0.40	1013	0.41
D	21.0	Creamy	585	0.17	1206	0.15
E	41.0	Good	449	0.27	1001	0.51
F	52.0	Good	411	0.32	984	0.43
G	42.0	Plastic	429	0.34	1021	0.42
H	50.0	Good	384	0.28	967	0.40
I	42.0	Good	499	0.15	1002	0.16

**Table 4 pharmaceutics-11-00175-t004:** Pycnometric densities, packing densities, packing index, porosity, and flowability of the experimental mini and conventional pellets.

Batch Code	Mini Pellets						Conventional Pellets					
	Ps (g/cc)	Pb (g/cc)	Pt (g/cc)	CC%	ε%	Flowability (g/s)	Ps (g/cc)	Pb (g/cc)	Pt (g/cc)	CC%	ε%	Flowability (g/s)
A	1.469	0.76	0.87	12.50	10.14	2.65	1.436	0.68	0.76	10.00	12.13	1.90
B	1.470	0.58	0.65	10.81	6.53	2.26	1.489	0.51	0.58	11.48	5.35	1.63
C	1.556	0.72	0.80	9.52	4.29	2.37	1.594	0.65	0.70	6.94	1.97	1.74
D	1.572	0.60	0.68	12.07	0.31	1.93	1.574	0.53	0.61	13.43	0.20	1.41
E	1.476	0.71	0.82	13.16	8.19	2.32	1.516	0.66	0.75	11.59	5.75	1.95
F	1.434	0.69	0.80	13.04	11.48	2.58	1.421	0.73	0.77	4.84	12.30	2.24
G	1.460	0.79	0.86	8.47	12.45	2.81	1.433	0.72	0.79	9.72	14.10	1.99
H	1.452	0.69	0.74	6.78	8.47	2.14	1.455	0.64	0.75	14.66	8.27	1.81
I	1.514	0.61	0.67	8.20	3.89	1.78	1.526	0.51	0.57	11.94	3.08	1.40

Ps: pycnometric density. Pb: bulk density. Pt: tap density. CC: Carr’s compressibility index. ε: porosity. Standard deviations for Ps < 0.015. Pb < 0.01. Pt < 0.01. CC% < 0.03 and for ε% < 0.83 (*n* = 3).

**Table 5 pharmaceutics-11-00175-t005:** Results of regression analysis of the experimental mixture design based on Scheffé quadratic models.

Response	Significance of Terms (*p*-Values)				Model Equation in Terms of Actual Components	*p*-sign.	R^2^	R_adj_^2^
	Linear Mixture	X_1_X_2_	X_1_X_3_	X_2_X_3_				
D_50_/mini	0.002	-	-	-	+3.45X_1_ + 4.67X_2_ + 7.10X_3_	0.002	0.837	0.742
D_50_/conv.	<0.001	0.004	0.058	0.006	+7.71X_1_ + 10.80X_2_ + 13.64X_3_	0.003	0.802	0.746
e_R_/mini	0.021	-	-	0.010	+2.79 × 10^−3^X_1_ + 3.12 × 10^−3^X_2_ −4.82 × 10^−3^X_3_ + 1.44 × 10^−4^X_2_X_3_	0.010	0.832	0.748
e_R_/conv.	0.036	-	-	0.120	+3.86 × 10^−3^X_1_ + 4.56 × 10^−3^X_2_ −7.37 × 10^−4^X_3_ + 1.21 × 10^−4^X_2_X_3_	0.042	0.721	0.581
Flowability/mini	0.015	0.273	0.163	-	+0.023X_1_ + 0.027X_2_ + 0.023X_3_ + 1.826 × 10^−4^X_1_X_2_ − 2.815 × 10^−4^X_1_X_3_	0.033	0.838	0.708
Flowability/conv.	<0.001	0.001	-	-	+0.013X_1_ + 0.019X_2_ + 0.013X_3_ + 3.518 × 10^−4^X_1_X_2_	<0.001	0.956	0.934
DE%/pH1.2/mini	0.298	-	0.012	-	+0.22X_1_ + 0.72X_2_ + 0.26X_3_ + 0.03X_1_X_3_	0.042	0.722	0.582
DE%/pH1.2/conv.	0.004	0.172	0.001	-	+0.07X_1_ + 0.72X_2_ + 0.44X_3_ + 5.61x10 − 3X_1_X_2_ + 0.02X_1_X_3_	0.002	0.945	0.902
DE%/pH5.6/mini	<0.001	0.084	0.015	0.001	+0.70X_1_ + 0.47X_2_ + 0.57X_3_ + 5.70x10^−3^X_1_X_2_ − 0.01X_1_X_3_ − 0.02X_2_X_3_	0.001	0.983	0.962
DE%/pH5.6/conv.	0.042	-	-	0.029	+0.16X_1_ +0.25X_2_ +0.71X_3_ −0.01X_2_X_3_	0.026	0.764	0.646
td/pH1.2/mini	0.543	0.072	0.009	-	1.44X_1_ + 0.702X_2_ + 1.188X_3_ − 0.025X_1_X_2_ − 0.054X_1_X_3_	0.039	0.825	0.685
td/pH1.2/conv.	0.069	-	0.042	-	1.079X_1_ + 0.439X_2_ + 0.676X_3_ − 0.030X_1_X_3_	0.044	0.717	0.576
td/pH5.6/mini	0.027	-	0.054	0.103	0.234X_1_ + 1.108X_2_ + 0.478X_3_ − 0.036X_1_X_3_ + 0.024X_2_X_3_	0.035	0.833	0.699
td/pH5.6/conv.	<0.001	-	<0.001	<0.001	0.084X_1_ + 0.6118X_2_ − 0.171X_3_ + 0.058X_1_X_3_ + 0.044X_2_X_3_	<0.001	0.979	0.963
%Released 2h/pH5.6	<0.001		0.016	0.002	0.817X_1_ + 0.513X_2_ + 0.570X_3_ − 0.01X_1_X_3_ − 0.02X_2_X_3_	<0.001	0.964	0.935
%Released 8h/pH5.6	0.002	0.029	-	0.002	0.737X_1_ + 0.790X_2_ + 0.747X_3_ + 0.013X_1_X_2_ − 0.025X_2_X_3_	0.002	0.953	0.916

X_1_: Chitosan. X_2_: Microcrystalline cellulose. X_3_: Drug. D_50_: median pellet diameter. e_R_: shape coefficient. DE%: dissolution efficiency. td: parameter in the Weibull equation.

**Table 6 pharmaceutics-11-00175-t006:** Crystallinity index (CI%) of the drug in the experimental pellet batches and the corresponding physical mixtures (PM). (Intensity of the drug at 25.8 2θ degrees 2132.55 a.u.).

Batch Code/Drug%	CI% of Drug in PM	CI% of Drug in Pellet	Crystallinity Loss (%)
B/5	24.73 ± 0.31	19.15 ± 0.05	22.56 ± 0.09
G/5	21.62 ± 0.02	17.05 ± 0.18	21.14 ± 0.17
A/21.25	35.96 ± 0.21	34.57 ± 0.15	3.87 ± 0.42
H/21.25	33.84 ± 0.03	32.99 ± 0.04	2.51 ± 0.48
C/37.5	55.29 ± 0.18	53.09 ± 0.16	3.98 ± 0.26
I/37.5	58.21 ± 0.19	56.73 ± 0.21	2.54 ± 0.34

**Table 7 pharmaceutics-11-00175-t007:** Dissolution efficiency (DE%) (mean ±SD, *n* = 3) and similarity factors (f2) comparing drug release from mini and conventional pellets at pH 1.2 and 5.6.

Batch Code	DE% for Mini Pellets at:		DE% for Conventional Pellets at:		Similarity Factor (f2) Comparing Mini and Conventional Pellets at:	
	pH 1.2	pH 5.6	pH 1.2	pH 5.6	pH 1.2	pH 5.6
A	67.56 ± 0.74	28.26 ± 0.17	69.48 ± 1.12	19.73 ± 0.39	77.90	51.93
B	38.85 ± 1.86	61.43 ± 1.52	30.44 ± 2.56	17.75 ± 1.89	44.44	37.00
C	44.87 ± 1.42	15.09 ± 1.05	58.53 ± 2.74	9.40 ± 1.14	50.83	61.14
D	64.46 ± 0.23	29.85 ± 0.68	70.16 ± 1.54	43.24 ± 0.27	62.07	53.94
E	70.70 ± 0.41	30.73 ± 0.37	71.01 ± 1.32	19.73 ± 0.84	49.30	47.75
F	43.78 ± 1.36	60.23 ± 1.89	52.44 ± 1.95	13.45 ± 1.07	52.48	31.74
G	60.64 ± 0.65	40.10 ± 0.41	66.85 ± 1.10	10.76 ± 0.31	69.82	45.43
H	70.41 ± 0.59	45.46 ± 1.24	67.03 ± 0.89	25.26 ± 0.27	41.26	35.77
I	71.84 ± 0.61	33.25 ± 1.18	73.05 ± 1.27	31.49 ± 0.02	53.02	65.12

**Table 8 pharmaceutics-11-00175-t008:** Weibull equation parameters (mean ±SD, n = 3) for the dissolution of mini and conventional pellets at pH 1.2 and 5.6.

Batch Code	Parameters at pH 1.2						Parameters at pH 5.6					
	Mini Pellets			Conventional Pellets			Mini Pellets			Conventional Pellets		
	*b* ^#^	td	R^2^	*b*	td	R^2^	*b*	td	R^2^	*b*	td	R^2^
A	1.16	40.3 ± 1.7	0.996	1.14	36.8 ± 1.5	0.993	0.73	116.2 ± 1.9	0.969	0.71	107.1 ± 1.2	0.990
B	1.05	81.3 ± 1.1	0.988	0.78	83.4 ± 2.6	0.982	0.69	42.3 ± 1.2	0.879	0.67	32.2 ± 0.7	0.968
C	1.15	79.2 ± 0.6	0.992	0.96	69.7 ± 1.9	0.985	0.70	121.1 ± 2.6	0.980	0.85	120.6 ± 1.9	0.992
D	0.74	26.9 ± 2.3	0.912	0.74	26.9 ± 2.9	0.912	0.76	105.9 ± 1.8	0.938	0.79	88.2 ± 2.1	0.994
E	0.96	26.9 ± 1.4	0.997	1.36	42.7 ± 3.5	0.981	0.75	127.9 ± 1.7	0.931	0.70	114.4 ± 1.7	0.963
F	0.69	41.9 ± 2.5	0.973	1.10	55.8 ± 2.7	0.987	0.74	73.8 ± 0.9	0.915	0.76	59.2 ± 1.1	0.956
G	1.12	56.1 ± 3.2	0.987	1.10	51.0 ± 2.3	0.989	0.63	111.2 ± 1.4	0.838	0.79	64.6 ± 0.3	0.972
H	0.70	18.7 ± 1.3	0.921	0.99	55.6 ± 2.3	0.991	0.69	112.2 ± 1.8	0.935	0.78	104.3 ± 1.3	0.970
I	0.81	19.5 ± 2.1	0.981	1.18	33.1 ± 1.8	0.981	0.69	96.1 ± 1.4	0.962	0.96	118.2 ± 1.8	0.991

^#^ For parameter *b*, SD < 0.01.

**Table 9 pharmaceutics-11-00175-t009:** Desirability values as computed by numerical optimization for fast (A) and extended release (B) pellets (Criteria for e_R_, Flowability, DE%, and td are described in the text. Drug content was set to target levels shown in the fourth column).

**A. Numerical Optimization Solutions for Instant Release**									
**Pellet Size**	**CHS2**	**MCC**	**Drug**	**e_R_**	**Flowability**	**DE%**	**td (pH 1.2)**	**Desirability**	
Mini Pellets	10.55	79.45	5.0	0.33	2.70	63.97	47.94	0.925	
	25.84	59.16	10.0	0.34	2.53	58.51	36.77	0.914	
	35.04	44.96	15.0	0.33	2.40	64.30	32.39	0.925	
	37.71	37.29	20.0	0.32	2.32	70.44	30.06	0.934	
	36.34	33.66	25.0	0.32	2.27	74.57	28.91	0.932	
	36.35	28.65	30.0	0.30	2.21	77.36	26.25	0.926	
	32.14	27.86	35.0	0.30	2.18	77.60	28.12	0.919	
	27.74	27.26	40.0	0.30	2.15	75.72	32.11	0.909	
Conventional	46.81	43.19	5.0	0.40	1.95	54.33	52.51	0.889	
	43.41	41.59	10.0	0.40	1.92	59.17	50.24	0.902	
	39.24	40.76	15.0	0.40	1.88	63.37	48.28	0.910	
	34.75	40.25	20.0	0.40	1.85	66.55	47.12	0.914	
	30.10	39.90	25.0	0.40	1.81	68.57	46.94	0.912	
	31.80	33.20	30.0	0.37	1.75	70.54	44.38	0.909	
	32.91	27.09	35.0	0.34	1.68	72.69	41.08	0.904	
	28.45	26.55	40.0	0.33	1.65	73.05	40.73	0.897	
**B. Numerical Optimization Solutions for Extended Release** Criteria: Shape index > 0.30. Flowability maximize. DE% at pH 1.2 maximize. Release in pH 5.6 in less than 2 h. Release in 5.6 at 8 h more than 70%. Drug content set to different target levels (column 4).									
**Mini Pellets**	**CHS2**	**MCC**	**Drug**	**e_R_**	**Flowability**	**DE%**	**Release (2 h)**	**Release (8 h)**	**Desirability**
	46.02	43.98	5.0	0.30	1.71	49.9	60.0	93.3	0.857
	47.32	37.68	10.0	0.33	1.61	54.0	53.1	85.4	0.726

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
