# Peer review of "Modulation of the Release of a Non-Interacting Low Solubility Drug from Chitosan Pellets Using Different Pellet Size, Composition and Numerical Optimization"

_pharmaceutics, 2019, doi:10.3390/pharmaceutics11040175_

Reviewer 1 Report

This is an excellent article about the preparation of chitosan pellets for piroxicam release by extrusion/spheronization. The article is well developed and write and I consider that can be accepted for publication in Pharmaceutics

Author Response

We thank the reviewer for the encouraging comments.

Reviewer 2 Report

Authors have reported optimization of the chitosan pellets of Piroxicam. The manuscript does not convey the novelty of the work. The manuscript is written in a language that is not easy to comprehend. Authors claimed that drug is cationic and cannot interact at higher pH but how can they defend the drug interaction at acidic pH with a cationic polymer such as Chitosan. Piroxicam causes serious damage to the stomach and Chitosan is highly soluble at acidic pH then why do we want to increase drug release or it solubility in this environment. Stay of the drug in the stomach is hardly 120 minutes, why the release studies in this pH were conducted up to 500 minutes. Why do the authors only use the Weibull model to predict the release mechanism? 
Data has not been presented statistically. The rationale for using d-optimal design is not present. 3d surface plots could have been more useful in visualizing the results. 

Author Response

Thank you for the comments.

Please find below our replies to the comments in the same order as they were cited.

Comment #1. Authors have reported optimization of the chitosan pellets of Piroxicam. The manuscript does not convey the novelty of the work.            

-The novelty of the work lies in the development of mini pellets (mean diameter below 0.5 mm) of a poorly soluble drug with chitosan. These provided complete and extended release over an 8-hour period when chitosan/drug ratio is ≥2 and drug content ≤ 21.25%. The development is supported in the manuscript by the results of physicochemical/technological characterization and by the release studies. The findings are cited in the Abstract and the Conclusions. They have not been previously reported and can be useful as a guide for the development of controlled release formulations of low solubility drugs. Therefore, they convey novelty.

#2. The manuscript is written in a language that is not easy to comprehend.

-Please advise what part of the manuscript is not easy to comprehend.

#3. Authors claimed that drug is cationic and cannot interact at higher pH but how can they defend the drug interaction at acidic pH with a cationic polymer such as Chitosan.

-There is not direct claim of such interaction in the manuscript. Τhe reviewer probably refers to lines 238-240 in the manuscript reading ‘…in the acidic environment the dissolved CHS chains form water-soluble units with the drug molecules attached to the chains, increasing considerably drug solubility’. A possible explanation for the formation of chitosan/drug water soluble units could be weak hydrogen bonding between hydroxyl groups present in chitosan with the hydroxyl of the protonated in the acidic pH zwitterionic tautomer of piroxicam (Ivanova et al 2015). Text has been added in lines 271-274 of revised.

#4. Piroxicam causes serious damage to the stomach and Chitosan is highly soluble at acidic pH then why do we want to increase drug release or its solubility in this environment.

-We agree with the possible damage of PXC to the stomach. However, co-administration with chitosan ameliorates this side effect since it is known that chitosan has potent cytoprotective and ulcer-healing action in gastric ulcers (reference 24 in the manuscript). Text has been added in the revised manuscript (lines 59, 60).

#5. Stay of the drug in the stomach is hardly 120 minutes, why the release studies in this pH were conducted up to 500 minutes.

-We agree that the stay of the drug in the stomach delivered by conventional formulations is hardly 120 min. However, due to the mucoadhesive properties of chitosan the gastric residence time may be increased considerably (Khatab and Zaki 2017, AAPS PharmSciTech, 18 (4), 2017). Therefore, we conducted the release studies up to 500 min since for some pellet batches release was extended up to this time. Text has been added in the revised manuscript (lines 490-494) together with supporting reference no 56 to explain this point.

#6. Why do the authors only use the Weibull model to predict the release mechanism?

-We decided to use this model as it presents a relatively newer kinetic approach that utilizes the entire drug release profile, and hence provides a more thorough description of the release mechanism (lines 81, 82 and references 29, 39 in the revised manuscript).

#7. Data has not been presented statistically.

-In response to this comment we have added standard deviations in:

Fig. 2 as error bars, Table 2 for moisture content and particle density, Table 4 at the end of the Table, Table 6 for the crystallinity index and crystallinity loss, Table 7 for the dissolution efficiency and in Table 8 for the parameters b, td.

#8. The rationale for using d-optimal design is not present.

-Since the influence of composition on the properties of pellets and drug release was studied for mini and conventional pellets separately, the number of experimental batches increased twice. For this reason, d-optimal mixture experimental design was employed as an efficient design applicable to the constrained design space used in this work (Myers and Montgomery, p. 590, 591). Text with the rationale has been added in the revised manuscript (lines 196-198) together with reference no 45.

#9. 3d surface plots could have been more useful in visualizing the results.

We think that contour plots provide clear graphical optimization of the effects of components on a studied property. Also, trace plots describe clearly the effect of a single component at a fixed ratio of the others. For this reason, we prefer to keep the presentation in Figures 3, 4, 5, 11, 12 as it is. However, if the reviewer insists, we will replace the contour with 3d surface plots. 

Reviewer 3 Report

The present manuscript deals with the optimization of formulation and drug release studies on chitosan-based pellets prepared using two different orifice sizes. Chitosan has been proposed as an excipient able to swell, differently from microcrystalline cellulose, and consequently, to slow drug release. The manuscript is well written and organised. However, for a clear investigation regarding the effect of chitosan on drug release, the comparison with the formulation without chitosan is required as already reported in the literature (only one example is “International Journal of Biological Macromolecules Volume 120, Part A, December 2018, Pages 1208-1215”). Please report the data also for the formulation without chitosan for comparison or, alternatively, discuss according to the available literature.

In figure 2b please report in the caption the concentration of chitosan used.

I think it is not correct to speak about drug solubility enhancing in the case of insoluble chitosan in deionised water. For an increase in solubility, the drug should be molecularly dispersed in the medium and not adsorbed onto insoluble fibers.

For deionised water, it is not possible to measure a reliable pH value. Despite sometimes, deionised water is referred in the literature having a pH of 5.6, this is not correct since it depends from the amount of dissolved CO2. I suggest using only deionised water without pH specification.

Line 242 and table 2:  the indicated values are not “Mean particle size” but “a particle size distribution according to percentile”. Please amend them.

Please implement the caption of all the tables with more information.

For a better reading and understanding, explain shortly in the manuscript how to interpret the results according to the trace plot, since it is not so commonly used.

Explain in the method section how crystallinity loss (%) values reported in table 6 have been calculated.

Line 408-409 Why the pH 5.6 is referred as neutral media??

Caption of figure 10 is not clear. Please rephrase.

Line 430-432 Please discuss more about the plateau at a drug release lower than 100% in drug release profile since this is a crucial point the release mechanism modellization.

Author Response

Thank you for the comments

Please find below our replies to the comments in the same order as they were cited.

Comment #1. The present manuscript deals with the optimization of formulation and drug release studies on chitosan-based pellets prepared using two different orifice sizes. Chitosan has been proposed as an excipient able to swell, differently from microcrystalline cellulose, and consequently, to slow drug release. The manuscript is well written and organized.

-We thank the reviewer for the comments.

#2. However, for a clear investigation regarding the effect of chitosan on drug release, the comparison with the formulation without chitosan is required as already reported in the literature (only one example is “International Journal of Biological Macromolecules Volume 120, Part A, December 2018, Pages 1208-1215”). Please report the data also for the formulation without chitosan for comparison or, alternatively, discuss according to the available literature.

-Data for drug release at pH 1.2 and 5.6 from mini and conventional pellets prepared without chitosan were already included in Figure 8 (black lines, open circles) of the original manuscript, where by mistake it was wrongly labelled as ’unprocessed drug’. This has now been corrected in the title and insert of Figure 8 in the revised manuscript. Discussion according to the available literature has also been added (lines 505-507 in the revised) together with a new reference no 57. 

#3. In figure 2b please report in the caption the concentration of chitosan used.

- The concentration of suspended chitosan was 100 mg in 10 ml deionized water. Text has been added in the caption of Figure 2 (lines 262-264) of revised manuscript.

#4. I think it is not correct to speak about drug solubility enhancing in the case of insoluble chitosan in deionised water. For an increase in solubility, the drug should be molecularly dispersed in the medium and not adsorbed onto insoluble fibers.

-Since solubility was determined from measurement of absorbance of the clear supernatant solutions following centrifugation, we think it is correct to speak about drug solubility. Drug adsorption onto the fibers aided wetting and dissolution (lines 268, 269) of revised.

#5 For deionised water, it is not possible to measure a reliable pH value. Despite sometimes, deionised water is referred in the literature having a pH of 5.6, this is not correct since it depends from the amount of dissolved CO2. I suggest using only deionised water without pH specification.

Deionised water was always kept in closed vessels. Additionally, during dissolution testing pH was measured in the beginning and in the end and there was no significant change. This has been clarified in the revised manuscript, lines 183-185.

#6 Line 242 and table 2:  the indicated values are not “Mean particle size” but “a particle size distribution according to percentile”. Please amend them.

-Thank you for the correction. The subheadings of Table 2 have been amended according to the reviewer and symbols explained at the end to the Table.

#7 Please implement the caption of all the tables with more information.

-Information has been added in the captions of Tables 1, 2, 3, 4, 6, 7 and 8, and explanatory footnotes at the end of Tables 2, 4 and 8.

#8 For a better reading and understanding, explain shortly in the manuscript how to interpret the results according to the trace plot, since it is not so commonly used.

-Explanation is provided in the revised manuscript (lines 215-219).

#9 Explain in the method section how crystallinity loss (%) values reported in table 6 have been calculated.

-Expression for the calculation of crystallinity loss (%) has been added in the revised as equation 1 (line 178) and the rest of the equations re-numbered accordingly.

#10 Line 408-409 Why the pH 5.6 is referred as neutral media??

-Thank you for the comment. Corrections have been made throughout the text (lines 33, 67, 245, 267, 441, etc).

#11  Caption of figure 10 is not clear. Please rephrase.

-Caption has been rephrased (lines 554, 555 of revised).

#12 Line 430-432 Please discuss more about the plateau at a drug release lower than 100% in drug release profile since this is a crucial point the release mechanism modellization.

-Thank you for the comment.

The plateau in Fig. 8a, b refers to the reported testing period of 8 h at pH 1.2. We conducted further measurements of drug release for longer time with the dissolution apparatus running overnight, which showed that release was still increasing, although slightly, compared to that at 8 h, reaching after 24 h between 85% and 90% for all mini pellet batches and also for conventional pellets except batch B (70% release). At pH 5.6 the %released after 24 h was between 80 and 95% for mini pellet batches except batch C (57%) and between 50% and 70% for conventional pellet batches except batch C (35% release). For the purpose of the present study the end of the test period was considered to be 8 h. Discussion has been added in the revised manuscript, lines 537-541.

Round  2

Reviewer 2 Report

The whole manuscript needs revision in terms of language editing.

Author Response

Comment #1 The whole manuscript needs revision in terms of language editing.

Response. The whole manuscript has been edited in terms of langauge as suggested by the reviewer.

Changes in the revised manuscript related to language editing can be seen using the 'Track changes' tool in Microsoft Word.

Text in the revised manuscript marked in red colour denotes changes requested by reviewer #3.

Reviewer 3 Report

#4. I think it is not correct to speak about drug solubility enhancing in the case of insoluble chitosan in deionised water. For an increase in solubility, the drug should be molecularly dispersed in the medium and not adsorbed onto insoluble fibers.

-Since solubility was determined from measurement of absorbance of the clear supernatant solutions following centrifugation, we think it is correct to speak about drug solubility. Drug adsorption onto the fibers aided wetting and dissolution (lines 268, 269) of revised.

I suggest changing the y-axis title in the figure 2 from “Solubility (mg/ml)” into “Apparent solubility (mg/ml)”, since this solubility value is not at thermodynamic equilibrium conditions. Columns in the graph of figure 2B have disappeared.

In table 1 under actual values (%), there is CXS2 instead of CHS2.  

#6 Line 242 and table 2:  the indicated values are not “Mean particle size” but “a particle size distribution according to percentile”. Please amend them.

-Thank you for the correction. The subheadings of Table 2 have been amended according to the reviewer and symbols explained at the end to the Table.

The authors should also change in the text (line 282) “mean particle size” with d50 percentile to be consistent and correct.

Author Response

Comment #1. 

I suggest changing the y-axis title in the figure 2 from “Solubility (mg/ml)” into “Apparent solubility (mg/ml)”, since this solubility value is not at thermodynamic equilibrium conditions. Columns in the graph of figure 2B have disappeared.

Response. 

Change has been made in Figure 2 and throughout the manuscript as suggested by the reviewer. Columns in Figure 2 can be seen now. 

Comment #2. 

In table 1 under actual values (%), there is CXS2 instead of CHS2. 

Response. 

CXS2 has been changed to CHS2 in Table 1 of revised manuscript.

Comment #3. 

The authors should also change in the text (line 282) “mean particle size” with d50 percentile to be consistent and correct.

Response

“mean particle size” has been changed to 'median (d50) particle diameter', in the revised,  line 289.  

Furthermore, the mansucript has been extensively edited in terms of English language.
